# Pressure Drop and Particle Settlement of Gas–Solid Two-Phase Flow in a Pipe

Wenqian Lin [1], Liang Li [2,3] and Yelong Wang [2,*]

[1] School of Media and Design, Hangzhou Dianzi University, Hangzhou 310018, China; lwq@hdu.edu.cn
[2] State Key Laboratory of Fluid Power Transmission and Control, Zhejiang University, Hangzhou 310027, China; liliang@zju.edu.cn
[3] Guangdong Infore Intelligent Sanitation Technology Co., Ltd., Foshan 528322, China
[*] Correspondence: wyl@zju.edu.cn

**Abstract:** Particle settlement and pressure drop in a gas–solid two-phase flow in a pipe with a circular cross-section are studied at mixture inlet velocities ($V$) ranging from 1 m/s to 30 m/s, particle volume concentrations ($\alpha_s$) ranging from 1% to 20%, particle mass flows ($m_s$) ranging from 5 t/h to 25 t/h, and particle diameters ($d_p$) ranging from 50 μm to 1000 μm. The momentum equations are based on a two-fluid model and are solved numerically. Some results are validated through comparison with the experimental results. The results showed that the gas and particle velocity distributions are asymmetrical around the center of the pipe and that the maximum velocity point moves up. The distance between the radial position of the maximum velocity and the center line for the gas is larger than that for the particles. The particle motion lags behind that of the gas flow. The particle settlement phenomenon is more serious, and the particle distribution on the cross-section is more inhomogeneous as the $V$, $\alpha_s$, and $m_s$ decrease and as $d_p$ increases. It can be divided into three areas according to the pressure changes along the flow direction, and the distinction between the three areas is more obvious as the $\alpha_s$ increases. The pressure drop per unit length increases as the $V$, $\alpha_s$ and $m_s$ increases and as $d_p$ decreases, Finally, the expressions of the settlement index and pressure drop per unit length as functions of $V$, $\alpha_s$, $m_s$, and $d_p$ are derived based on the numerical data.

**Keywords:** gas–solid two-phase flow; pipe with circular cross-section; settlement; pressure drop; numerical simulation

## 1. Introduction

Particle transport through a pipe is quite common in the power generation, metallurgy, machinery manufacturing, pharmaceutical and food production, and material engineering industries, among others. In transport processes, it is important to characterize the pressure drops and particle settlement [1–4], which are directly related to the transport efficiency and particle deposition to the wall-even blockage, well.

Some research has already been published on particle settlement and pressure drops in gas–solid two-phase flow in a pipe. Tong et al. [5] showed that vortex shedding resulting from natural convection changed the sedimentation velocity and induced horizontal oscillation. Balakin et al. [6] performed a study on particle sedimentation in suspensions with high particle concentrations and pointed out that Eulerian–Eulerian simulations could account for some of the detailed particle-settling processes. Tao et al. [7] found that the initial geometric arrangement of multiple particles had a great effect on sedimentation behavior. Chiodi [8] indicated that the transport of dense particles depended on the ratio of the shear velocity of the flow to the settling velocity of the particles and the Reynold's number of the sedimentation. Senapati and Dash [9] reported that the pressure drops showed completely opposite trends in two situations with different particle concentrations were used. The pressure drops increased as the particle volume concentration increased. Naveh et al. [10] found that the pressure-drop increase rate depended strongly on the Archimedes number.

Ariyaratne et al. [11] indicated that at higher gas velocities, the pressure drops predicted using the standard $k$-$\omega$ turbulence model are higher than those obtained when using the standard $k$-$\varepsilon$ model. Narimatsu and Ferreira [12] presented the minimum pressure gradient point experimentally through pressure gradient versus gas velocity curves and indicated that the transition velocity between dense and diluted flows enhanced as the particle density and diameter increased. Herbreteau and Bouard [13] presented a pressure drop- and Froude number-dependence on the particle size and density.

As shown above, although there have been some studies on particle settlement and pressure drop in gas–solid two-phase flow, few studied both at the same time. In addition, the factors affecting particle settlement and pressure drop include inlet velocity, particle volume concentration, particle mass flow, particle diameter, and so on, but there is no correlation expression between particle settlement, pressure drop, and these factors. Therefore, in the present study, the momentum equations based on a two-fluid model are solved numerically, and the distributions of velocity and particle concentration as well as pressure drop are analyzed. The effects of inlet velocity, particle volume concentration, particle mass flow, and particle diameter on particle settlement and pressure drop are discussed. Finally, the relationship between the settlement index, pressure drop, and related synthetic parameters is determined based on the numerical data.

## 2. Basic Equations

Figure 1 shows gas–solid two-phase flow in a pipe with diameter $D$ and length $L$. A two-fluid model is used to simulate three-dimensional gas–solid two-phase flow [14]. The particle phase is also considered to be a continuous medium in the two-fluid model, so particle-to-particle interaction has been reflected by the relationship between the stress and strain rates in the second term on the left-hand side of Equation (2). The two phases are regarded as two interacting continuous phases for the model, so the two phases have the same structure of the governing equations. Assuming that the flow field is steady and isothermal, there is no mass exchange between phases, and the particle stress tensor is ignored. Then, the continuity equation, momentum equation, and state equation are:

$$\nabla \cdot \left( \alpha_{g,s} \rho_{g,s} v_{g,s} \right) = 0, \tag{1}$$

$$\nabla \left[ \alpha_{g,s} \rho_{g,s} v_{g,s} v_{g,s} + \alpha_{g,s} \Gamma_{g,s} \left( \nabla v_{g,s} + \nabla v_{g,s}^T \right) \right] = -\nabla \left( \alpha_{g,s} p \right) + \alpha_{g,s} \rho_{g,s} g + M_{g,s}, \tag{2}$$

$$p = \rho_g R T_g, \tag{3}$$

where subscript "$g$, $s$" indicates the gas or solid phase; $\alpha$ is the phase composition; $\rho$ is the density; $v$ is the velocity; $\Gamma = \rho(v_l + v_t)$ is the diffusion coefficient; $v_l$ and $v_t$ are the molecular and turbulent viscosity coefficients, respectively; $p$ is the pressure; $g$ is the gravitational acceleration; $R$ is the gas constant; $T$ is the temperature; and $M$ is the interphase momentum exchange term:

$$M_{g,s} = K \left( v_{g,si} - v_{g,s} \right) + p_{g,s} \nabla \alpha_{g,s}, \tag{4}$$

where subscript "$i$" indicates the different phases, and $K$ is the interphase friction coefficient and can be expressed as follows when the particle volume concentration is larger than 0.2:

$$K = 150 \frac{\alpha_s^2}{\alpha_g} \frac{\mu}{d_s^2} + 1.75 \alpha_g \frac{1}{d_s} \rho_g |v_g - v_s|, \tag{5}$$

where $d$ is the particle diameter, and $\mu$ is the gas viscosity coefficient. When the particle volume concentration is less than 0.2, $K$ can be expressed based on the aerodynamic force acting on solid particles as follows:

$$K = \left( C_D \alpha_g^{-2.65} \right) \left( \frac{3 \alpha_s}{2 d_s} \right) \frac{1}{2} \alpha_g \rho_g |v_g - v_s|, \tag{6}$$

where the drag coefficient of a single particle $C_D$ and particle's Reynolds number $Re$ are:

$$C_D = Max\left\{\frac{24}{Re}\left(1 + 0.15Re^{0.687}\right), 0.44\right\}, \; Re = \frac{\rho_g d_s\left(\alpha_g|v_g - v_s|\right)}{\mu} \tag{7}$$

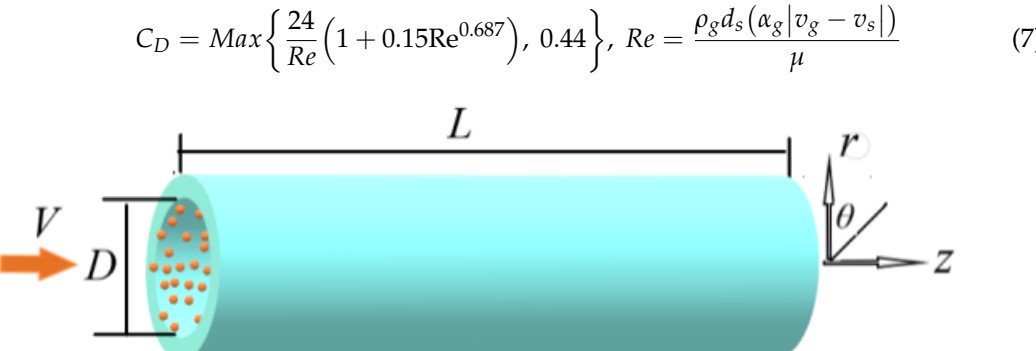

**Figure 1.** Schematic diagram of gas–solid two-phase flow in a pipe.

In the computation, the turbulent stress in the momentum equation adopts the Buossinesq eddy viscosity model, and the value of the eddy viscosity is determined with the corrected $k$-$\varepsilon$ turbulence model.

The friction between the gas and the pipe wall can be represented by adding a source phase to the gas phase momentum equation of the control body near the pipe wall. Assuming that there is no slip between the flow and the wall, the velocity near the wall is distributed logarithmically, and the wall friction is calculated according to the smooth pipe. The friction between the solid phase and the pipe wall can be calculated by the method in reference [15], i.e., a source phase is added to the momentum equation of the solid phase of the control body near the tube wall.

### 3. Numerical Simulation

*3.1. Parameters*

The IPSA_FULL method was used to numerically simulate Equations (1)–(7). IPSA refers to the Inter-phase Slip Algorithm, and FULL refers to the full elimination algorithm, which is full coupled with the implicit approach used in Flent-4.5. This method has been proven to significantly enhance the convergence of the numerical scheme [16].

On the wall, the velocity of the gas and particles satisfies the no-slip condition. The related parameters in the simulation are particle diameter $d_p$ = 50 μm, 100 μm, 500 μm, 750 μm, 1000 μm; mixture inlet velocity $V$ = 1 m/s, 7 m/s, 15 m/s, 23 m/s, 30 m/s; particle volume concentration $\alpha_s$ = 1%, 5%, 10%, 15%, 20%; particle mass flow $m_s$ = 5 t/h, 10 t/h, 15 t/h, 20 t/h, 25 t/h; gas density $\rho_g$ = 1.189 kg/m³; gas viscosity $v$ = 1.5440 × 10$^{-5}$ m²/s; the pipe outlet is 1 atmospheric pressure.

*3.2. Validation*

The grid system was composed of 32($r$) × 32($\theta$) × 208($z$) = 212992 grid points. Grid independence was tested by changing the values of the grid points from 24 to 40, 24 to 40, and 192 to 224 in the $r$, $\theta$, and $z$ directions, respectively. Table 1 shows the tested results, where a convergence criterion is specified with all of the residual errors being less than 10$^{-4}$.

**Table 1.** Values of $\Delta p/L$ when changing grid points.

| $r \times \theta \times S$ | $\Delta p/L$ | $r \times \theta \times S$ | $\Delta p/L$ | $r \times \theta \times S$ | $\Delta p/L$ |
|---|---|---|---|---|---|
| 24 × 32 × 208 | 50,891 | 32 × 24 × 208 | 50,889 | 32 × 32 × 192 | 50,893 |
| 28 × 32 × 208 | 50,880 | 32 × 28 × 208 | 50,879 | 32 × 32 × 200 | 50,881 |
| 32 × 32 × 208 | 50,872 | 32 × 32 × 208 | 50,872 | 32 × 32 × 208 | 50,872 |
| 36 × 32 × 208 | 50,868 | 32 × 36 × 208 | 50,869 | 32 × 32 × 216 | 50,867 |
| 40 × 32 × 208 | 50,865 | 32 × 40 × 208 | 50,867 | 32 × 32 × 224 | 50,864 |

In order to verify the numerical method and program used in the gas–solid two-phase flow simulation, we compared the present numerical results with the previous results [17], as shown in Figures 2 and 3, where $\alpha_{sa}$ and $v_{gza}$ are the average particle volume concentration and average gas velocity on the cross-section, respectively. We can see that the present numerical results and experimental results are qualitatively consistent.

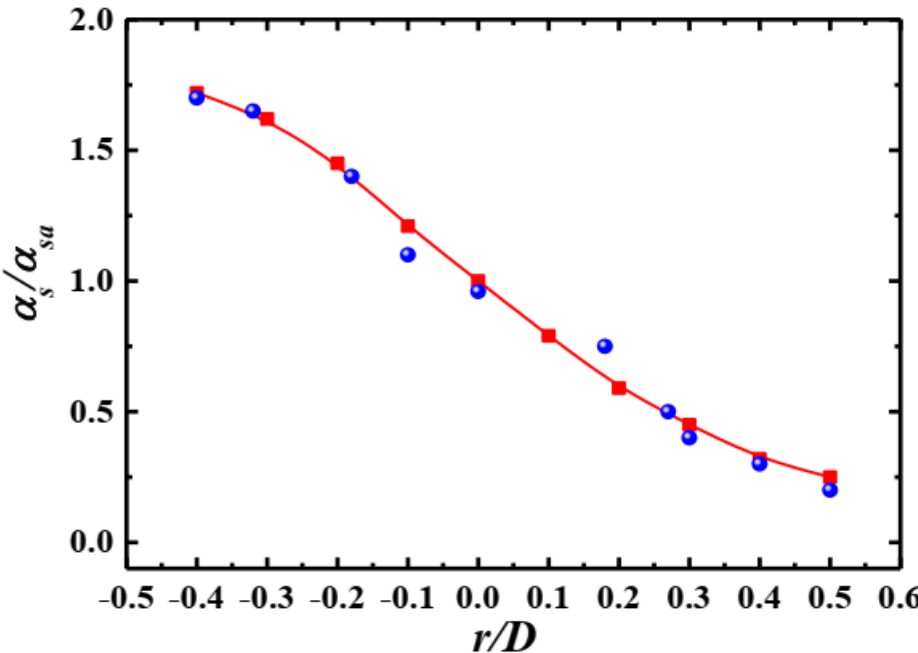

**Figure 2.** Distribution of particle concentration along the radial direction ($\alpha_s = 5\%$, $V = 7$ m/s). ■: present results; ●: experimental results [17].

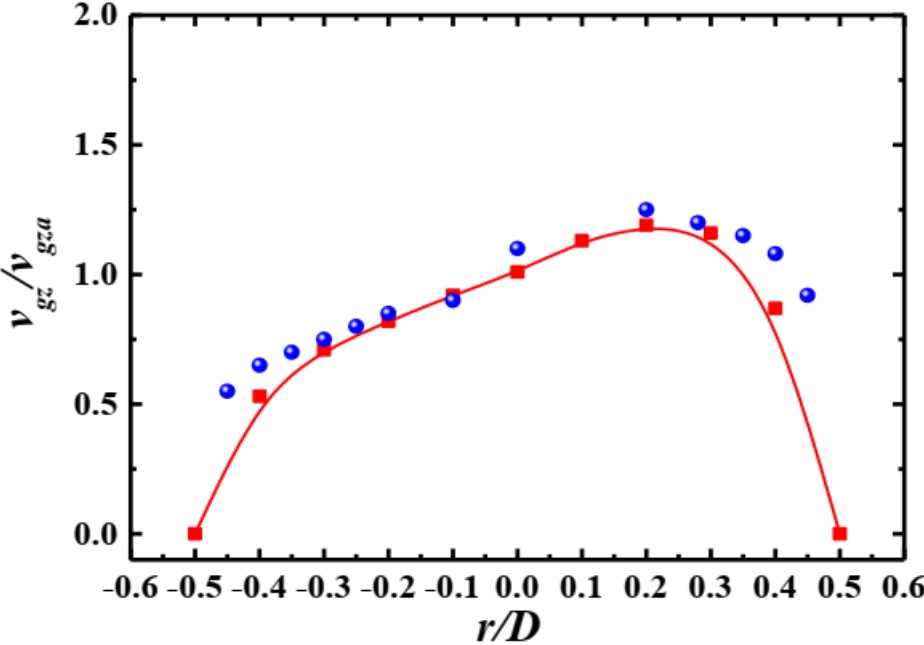

**Figure 3.** Distribution of gas velocity along the radial direction ($\alpha_s = 5\%$, $V = 7$ m/s). ■: present results ●: experimental results [17].

## 4. Results and Discussion

### 4.1. Distribution of Pressure along the Flow Direction

The pressure distributions along the flow direction for different particle volume concentrations are shown in Figure 4, where the pressure values are relative to the atmospheric pressure at the pipe outlet. It can be seen that four curves have the same change trend. The pressure at the inlet is the maximum, and it then decreases gradually to atmospheric pressure at the outlet because it is the flow caused by the pressure difference between the inlet and outlet. It can be divided into three areas according to the changes in the pressure. The pressure is high and changes slowly in the inlet area ($0 \leq z/L \leq 0.2$). The pressure change begins to increase in the transition area ($0.2 \leq z/L \leq 0.8$). The pressure decreases approximately linearly as the pipe length increases in the fully developed area ($0.8 \leq z/L \leq 1$) where the pressure drop per unit length is a constant that increases as the particle volume concentration increases. The distinction between the three areas is more obvious as the particle volume concentration increases.

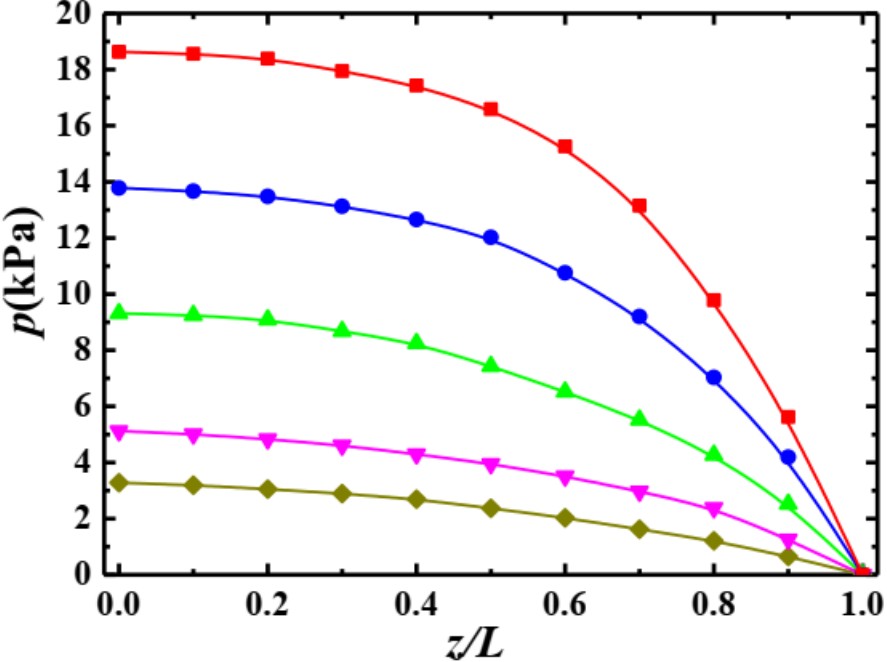

**Figure 4.** Pressure distributions along the flow direction ($d_p$ = 100 μm, $V$ = 7 m/s, $m_s$ = 15 t/h). ■: $\alpha_s$ = 20%; ●: $\alpha_s$ = 15%; ▲: $\alpha_s$ = 10%; ▼: $\alpha_s$ = 5%; ◆: $\alpha_s$ = 1%.

### 4.2. Distribution of Velocity along the Flow Direction

Figure 5 shows the velocity distribution of the gas and solid phases along the flow direction of the pipe. We can see that unlike single-phase flow, the gas velocity distribution is asymmetrical around the center of the pipe and that the maximum velocity point moves up because the particles are gradually moved to the lower part of the pipe by gravity, obstructing the motion of the lower gas flow, thus decreasing the lower gas velocity and increasing the upper gas velocity. The particle velocity distribution is similar to that of gas flow, and the difference is that the velocity is smaller. The gas and particle velocities are distributed uniformly at the inlet and show a parabolic velocity profile at $z/L$ = 0.2. Then, the velocity profile changes continuously along the flow direction until $z/L$ = 0.8, where the flow reaches a fully developed stable state. From the inlet area to the fully developed area, the gas velocity increases slightly, and the velocity profile becomes asymmetric around the center, and the overall gas velocity is higher than that of the particles. The maximum gas velocity occurs at $r/D$ = 0.35, while the maximum particle velocity appears near $r/D$ = 0.22. The distance between the radial position of the maximum velocity and the center line for

the gas is larger than that for the particles. The particle motion lags behind that of the gas flow.

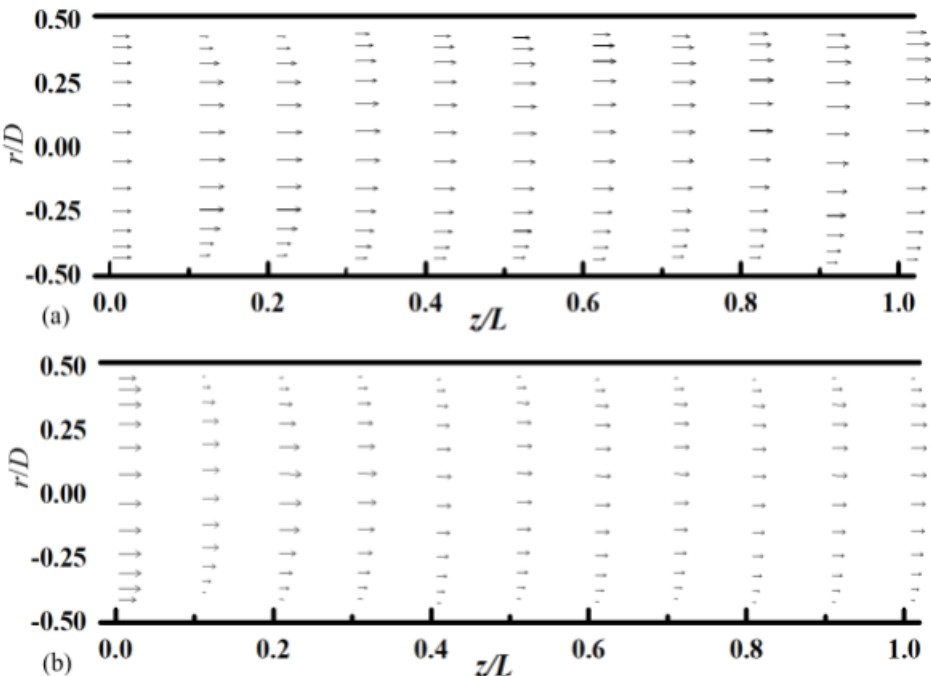

**Figure 5.** Velocity vector distribution at different sections ($d_p$ = 100 μm, $V$ = 7 m/s, $\alpha_s$ = 5%, $m_s$ = 15 t/h). (**a**) gas phase; (**b**) solid phase.

*4.3. Distribution of Particle Volume Concentration*

Particle volume concentration distributions on different cross-sections along the flow direction are shown in Figure 6, where the darker the color, the higher the concentration. The particle volume concentration is distributed uniformly at the inlet. As the flow develops downstream, the particle volume concentration increases and decreases gradually at the bottom and the upper part of the pipe, respectively, showing obvious particle sedimentation.

*4.4. Relationship between the Settlement Index and Mixture Inlet Velocity*

On the cross-section at the outlet, we can define a dimensionless settlement index:

$$Se = \frac{\alpha_{sb} - \alpha_{su}}{\alpha_{si}}, \tag{8}$$

where $\alpha_{sb}$, $\alpha_{su}$, and $\alpha_{si}$ are the particle volume concentration near the lower wall ($l/D$ = 0.125 as shown in Figure 6), near the upper wall ($l/D$ = 0.125), and at the inlet, respectively. *Se* indicates the sedimentation degree of the particles. The larger the value of *Se*, the larger the particle volume concentration difference near the upper and lower walls, i.e., the more obvious the particle settlement.

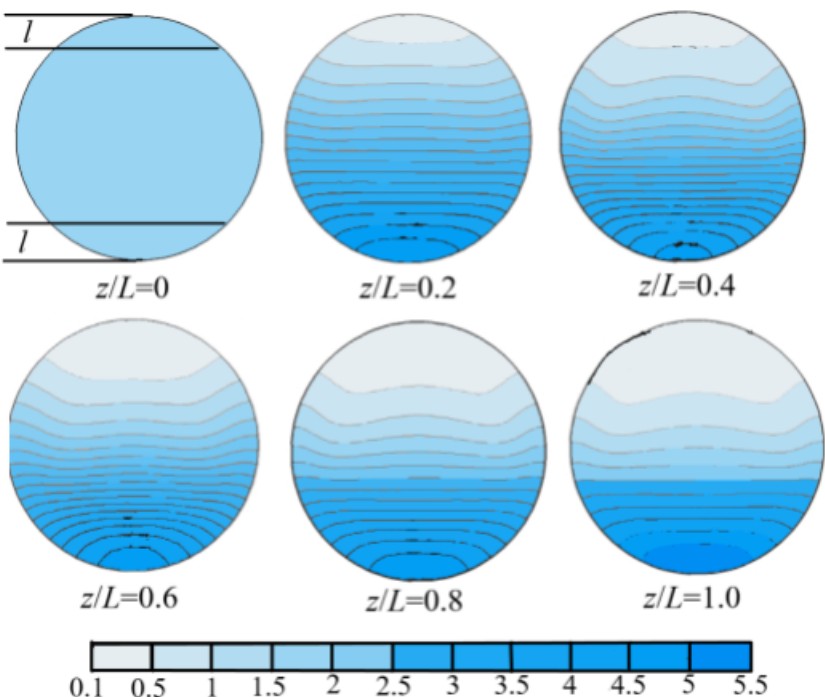

**Figure 6.** Particle volume distributions on different cross-sections along the flow direction ($d_p$ = 100 μm, $V$ = 7 m/s, $\alpha_s$ = 5%, $m_s$ = 15 t/h).

### 4.4.1. Effect of Particle Volume Concentration

Figure 7 shows the relationship between the settlement index $Se$ and mixture inlet velocity $V$ for different particle volume concentrations. It can be seen that $Se$ increases as $V$ decreases, i.e., the particle settlement phenomenon is more serious and the particle distribution on the cross-section is more inhomogeneous with the decrease in the mixture inlet velocity. The reason for this is that the conveying time that the particles spend in the pipe is longer at a small inlet velocity, so the settling time due to gravity is longer. $Se$ decreases as the $\alpha_s$ increases, which can be seen in the figure. This is partly due to the fact that the value of $Se$ is inversely proportional to $\alpha_s$, as shown in expression (8); on the other hand, a high particle volume concentration will hinder particle settlement.

### 4.4.2. Effect of Particle Mass Flow

The relationship between the settlement index $Se$ and mixture inlet velocity $V$ for different particle mass flow rates is shown in Figure 8, where it can be seen that the value of $Se$ decreases as the $V$ and particle mass flow $m_s$ increase. Actually, the particle mass flow is proportional to the particle volume flow when the particle density remains unchanged, while the particle volume flow is proportional to the volume concentration within a fixed time. Therefore, the principle of $Se$ increasing with $m_s$ is the same as that in Figure 7.

### 4.4.3. Effect of Particle Diameter

Figure 9 shows the relationship between the settlement index $Se$ and mixture inlet velocity $V$ for different particle diameters. It can be seen that $Se$ increases as the particle diameter $d_p$ increases. It is obvious that the larger the particle diameter, the more significant the particle settlement is, resulting in the particles having a more inhomogeneous distribution on the cross-section at the outlet. In addition, under the parameters considered in the present study, the values of $Se$ are larger in the case of different particle diameters than they are that in other cases, which shows that the particle diameter has a more significant effect on the uniformity of the particle distribution on the cross-section at the outlet.

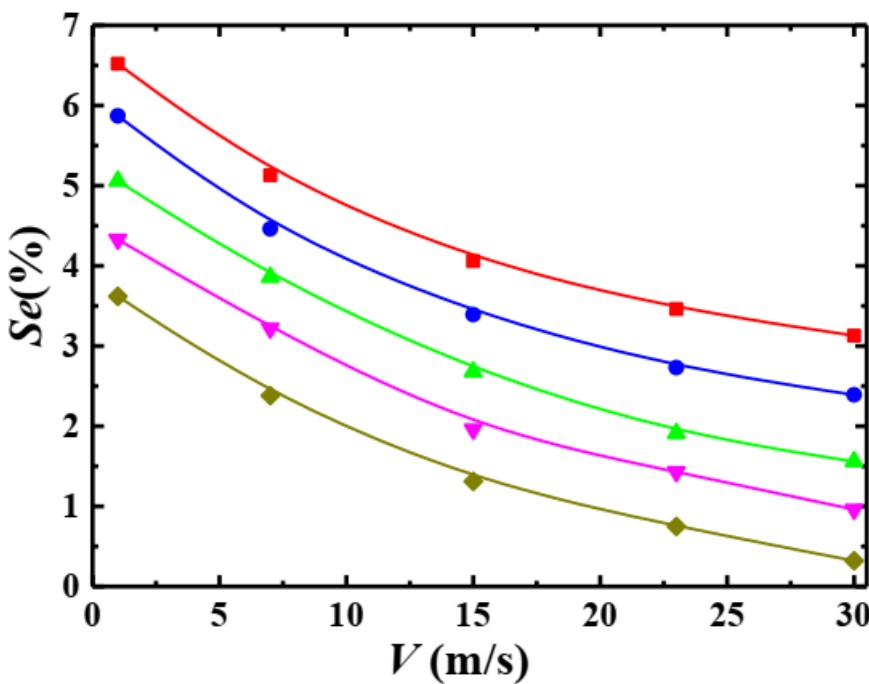

**Figure 7.** Relationship between *Se* and *V* for different concentrations ($d_p$ = 100 μm, $m_s$ = 15 t/h). $\alpha_s$: ■: 1%; ●: 5%; ▲: 10%; ▼: 15%; ♦: 20%.

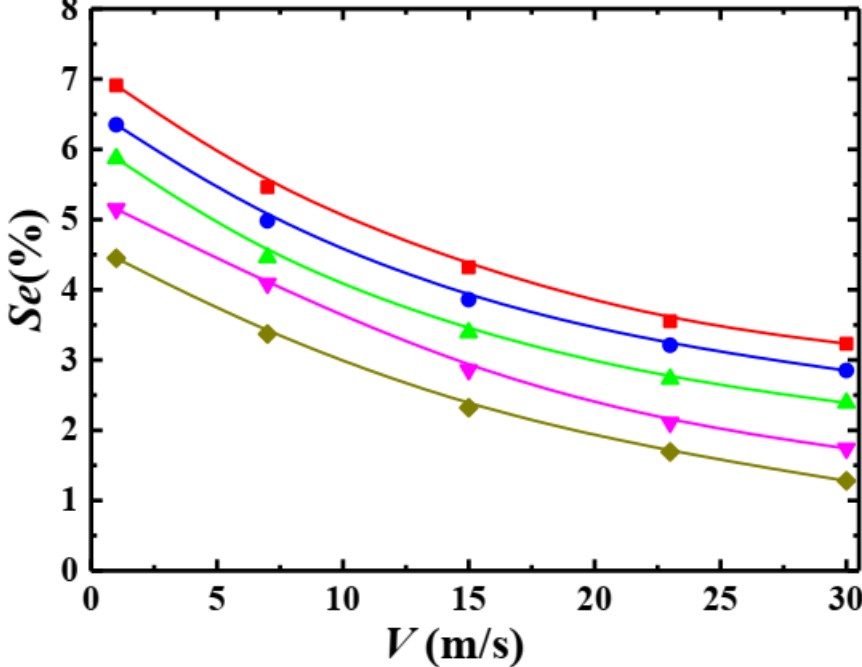

**Figure 8.** Relationship between *Se* and *V* for different particle mass flow ($d_p$ = 100 μm, $\alpha_s$ = 5%). $m_s$: ■: 5 t/h; ●: 10 t/h; ▲: 15 t/h; ▼: 20 t/h; ♦: 25 t/h.

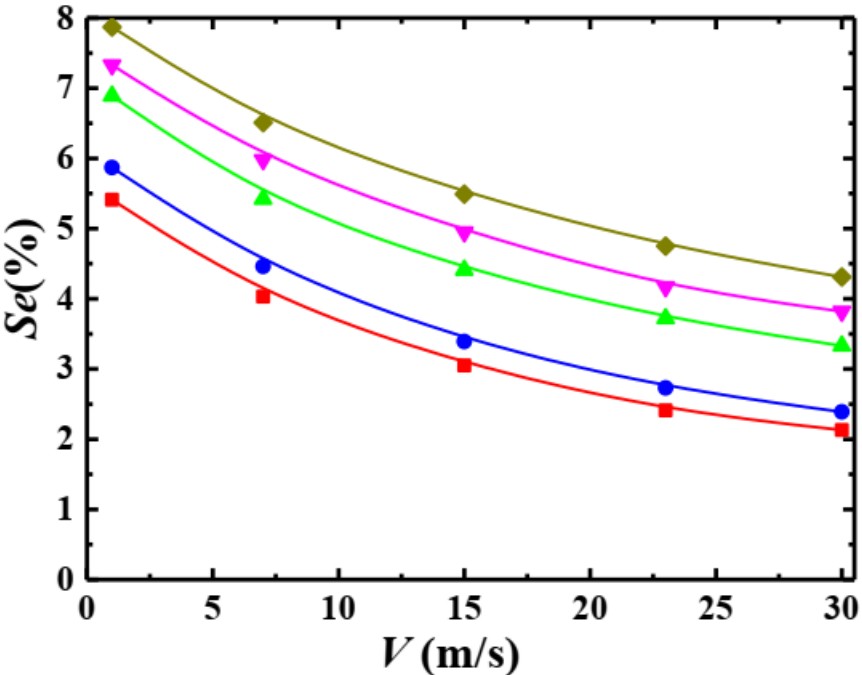

**Figure 9.** Relationship between *Se* and *V* for different particle diameters ($\alpha_s$ = 5%, $m_s$ = 15 t/h). $d_p$: ■: 50 μm; ●: 100 μm; ▲: 500 μm; ▼: 750 μm; ◆: 1000 μm.

*4.5. Relationship between the Pressure Drop and Mixture Inlet Velocity*

4.5.1. Effect of Particle Volume Concentration

The relationship between the pressure drop per unit length $p/L$ and mixture inlet velocity *V* for different particle volume concentrations is shown in Figure 10, where the values of $p/L$ increase as the *V* increases for different volume concentrations, which is in accordance with the law that the pressure drop is directly proportional to the velocity in the Hagen–Poiseuille flow. In the figure, the values of $p/L$ also increase as the particle volume concentration increases for the different inlet velocities. Since the particle density is larger than that of the gas density, the high particle concentration per unit volume means that a larger pressure drop is required to transport the mixture over the same distance.

4.5.2. Effect of Particle Mass Flow

Figure 11 shows the relationship between the pressure drop per unit length $p/L$ and mixture inlet velocity *V* for different particle mass flows. For a given inlet velocity, the values of $p/L$ increase as the particle mass flow increases because the particle mass flow is directly proportional to the particle volume concentration.

4.5.3. Effect of Particle Diameter

The relationship between the pressure drop per unit length $p/L$ and mixture inlet velocity *V* for different particle diameters is shown in Figure 12, where we can see that the values of $p/L$ decrease as the particle diameter increases. The reason for this is that under conditions where the particle volume concentration is constant, the smaller the particle, the more particles there are, and the stronger the effect of particle gas interaction. The stronger particle and gas interactions makes the required pressure drop larger due to the drag effect of the gas on the particles. It is obvious that the effect of particle diameter on pressure drop is less significant than that of the particle volume concentration and mass flow.

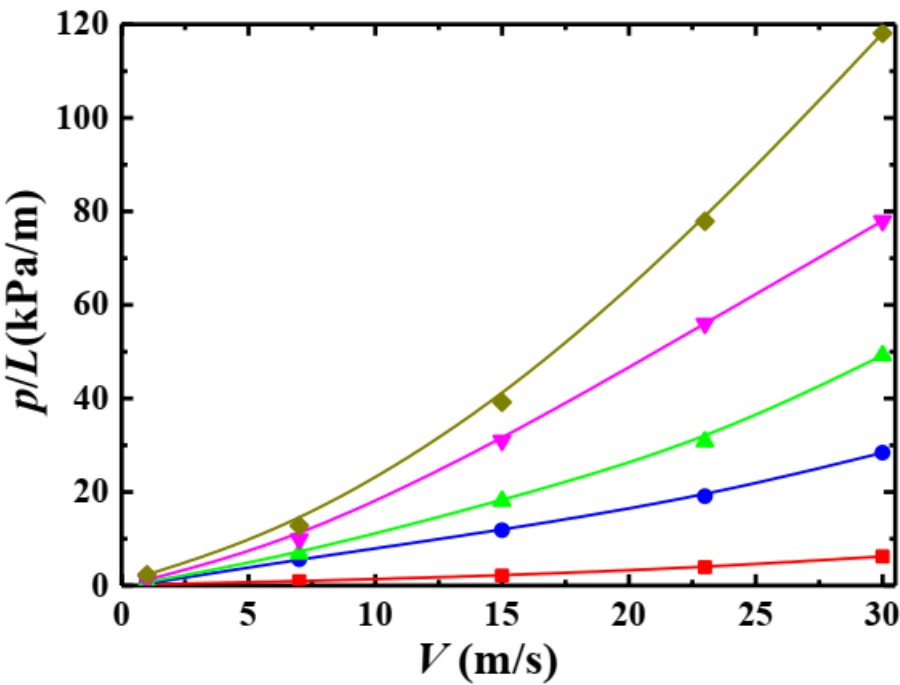

**Figure 10.** Relationship between $p/L$ and $V$ for different concentrations ($d_p$ = 100 μm, $m_s$ = 15 t/h). $\alpha_s$: ■: 1%; ●: 5%; ▲: 10%; ▼: 15%; ◆: 20%.

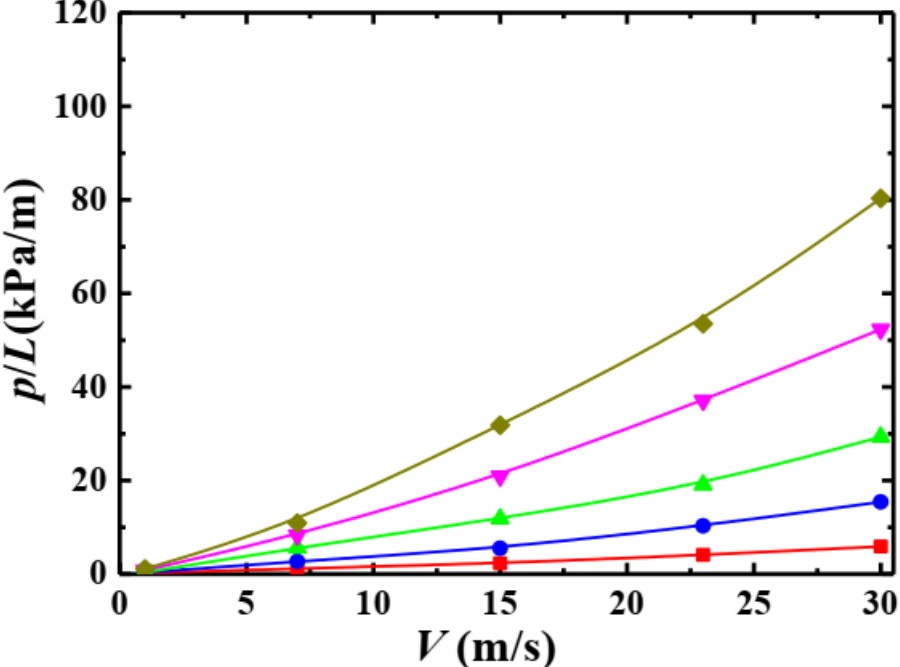

**Figure 11.** Relationship between $p/L$ and $V$ for different particle mass flow ($d_p$ = 100 μm, $\alpha_s$ = 5%). $m_s$: ■: 5 t/h; ●: 10 t/h; ▲: 15 t/h; ▼: 20 t/h; ◆: 25 t/h.

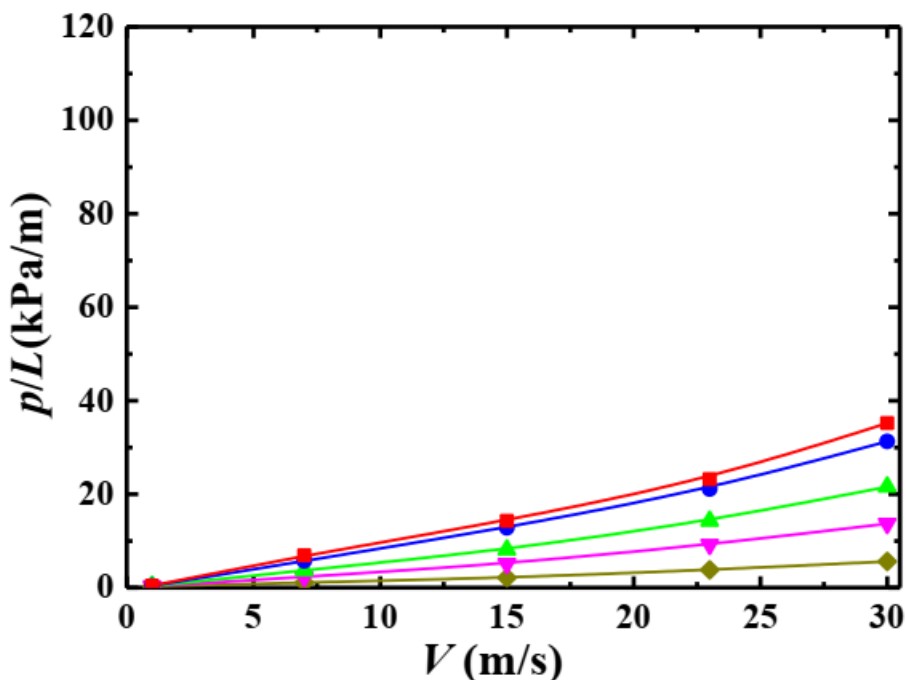

**Figure 12.** Relationship between $p/L$ and $V$ for different particle diameters ($\alpha_s$ = 5%, $m_s$ = 15 t/h). $d_p$: ■: 50 μm; ●: 100 μm; ▲: 500 μm; ▼: 750 μm; ◆: 1000 μm.

*4.6. Relationship of Settlement Index, Pressure Drop and Related Parameters*

It is necessary to build a relationship between the settlement index, pressure drop per unit length, and related parameters in order to effectively characterize the gas–solid two-phase flow in a pipe. As shown in Figures 7–9, the settlement index $Se$ is inversely proportional to the inlet velocity $V$, particle volume concentrations $\alpha_s$, and particle mass flow $m_s$, while it is also directly proportional to the particle diameter $d_p$. As shown in Figures 10–12, the pressure drop per unit length $p/L$ is proportional to the inlet velocity $V$, particle volume concentrations $\alpha_s$, and particle mass flow $m_s$, while it is inversely proportional to the particle diameter $d_p$. As such, we can combine $V$, $\alpha_s$, $m_s$, and $d_p$ into a synthetic parameter:

$$\eta = \frac{d_p}{V \alpha_s m_s}, \quad \zeta = \frac{V \alpha_s m_s}{d_p}. \tag{9}$$

Based on the above numerical data and expression (9), we can establish the following settlement index $Se$ and pressure drop per unit length $p/L$ formula:

$$Se = 0.69854 + 2.68062 \log \eta - 0.15859 (\log \eta)^2, \tag{10}$$

$$\frac{p}{L} = 119.33207 \exp\left(\frac{\zeta}{1.35994}\right) - 117.66952. \tag{11}$$

The settlement index and pressure drop per unit length as a function of related synthetic parameter are shown in Figures 13 and 14, where each solid dot represents different numerical data under different $\eta$ and $\zeta$, which are composed of different $V$, $\alpha_s$, $m_s$, and $d_p$.

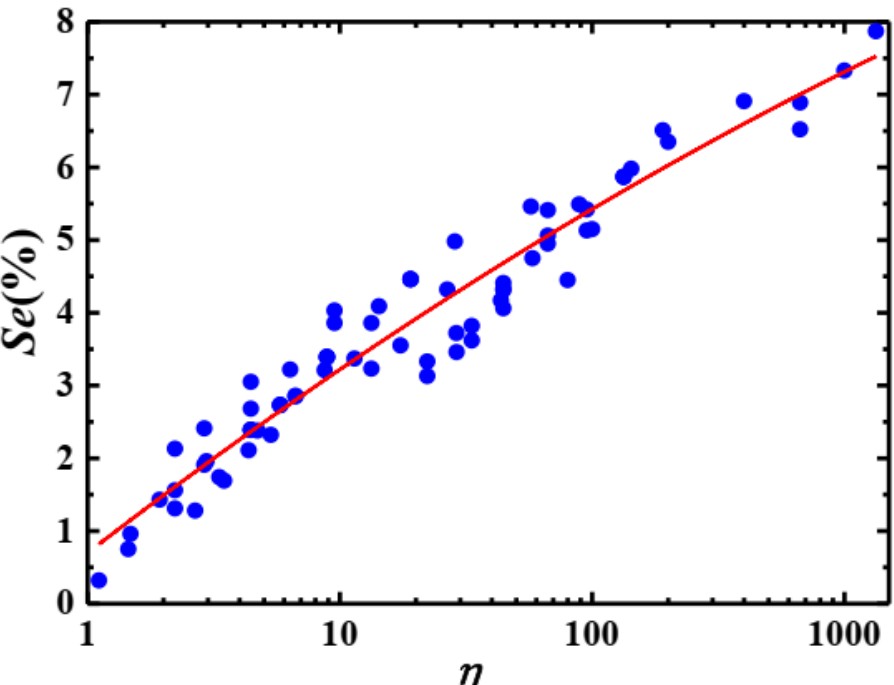

**Figure 13.** Relationship between *Se* and *η*. ●: numerical data; ——: Formula (10).

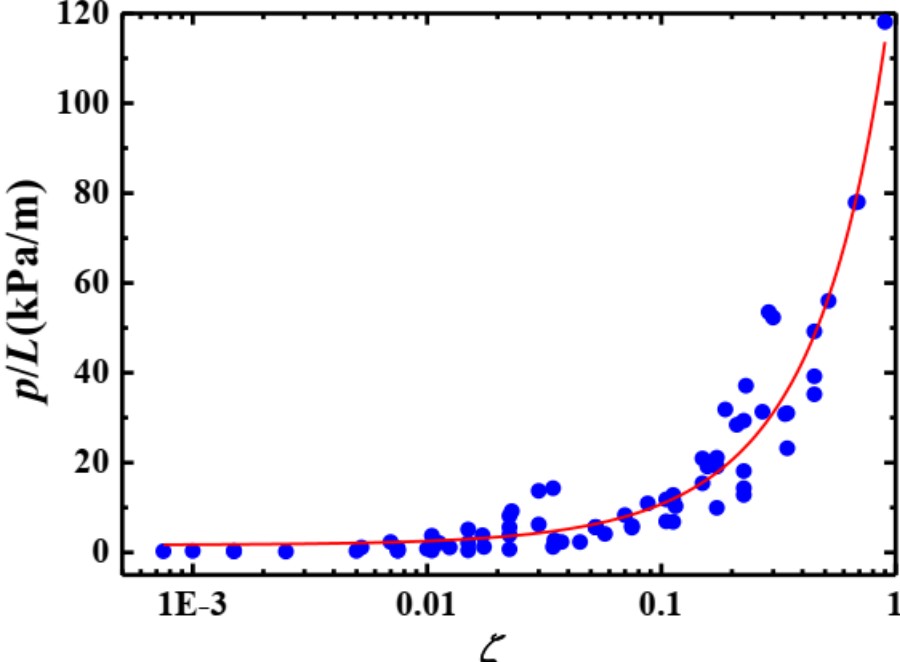

**Figure 14.** Relationship between *Se* and *ζ*. ●: numerical data; ——: Formula (11).

## 5. Conclusions

In order to clarify the effect of inlet velocity, particle volume concentration, particle mass flow, and particle diameter on the sedimentation degree of particle and pressure drop in gas–solid two-phase flow in a pipe with a circular cross-section, the continuity equation, momentum equation, and state equation in the range of mixture inlet velocities ranging from 1 m/s to 30 m/s, particle volume concentrations ranging from 1% to 20%, particle mass flows ranging from 5 t/h to 25 t/h, and particle diameters ranging from 50 μm to 1000 μm were solved numerically based on a two-fluid model. Some results were

validated by comparing the experimental results. The main conclusions can be summarized as follows:

(1) The gas and particle velocity distributions are asymmetrical around the center of the pipe and as the maximum velocity point moves up. The distance between the radial position of the maximum velocity and the center line for the gas is larger than that for the particles. The particle motions lags behind that of the gas flow.

(2) As the flow develops downstream, the particle volume concentration increases and decreases gradually at the bottom and the upper part of the pipe, respectively, showing obvious particle sedimentation. The particle settlement phenomenon is more serious, and the particle distribution on the cross-section is more inhomogeneous with the decrease in the mixture inlet velocity, particle volume concentration, and particle mass flow as well as with the increase in the particle diameter.

(3) The pressure at the inlet is the maximum, and it then decreases gradually to the atmospheric pressure at the outlet. It can be divided into three areas according to the pressure changes, i.e., inlet area, transition area, and fully developed area. The distinction between the three areas is more obvious as the particle volume concentration increases. The pressure drop per unit length increases as the mixture inlet velocity, particle volume concentration, and particle mass flow increase and as the particle diameter decreases.

(4) Finally, the expressions of settlement index and pressure drop per unit length functions of the mixture inlet velocity, particle volume concentration, particle mass flow, and particle diameter are derived based on numerical data in order for the settlement index and pressure drop to be calculated conveniently.

**Author Contributions:** Conceptualization, Y.W. and W.L.; methodology, W.L. and L.L.; software, W.L. and L.L.; validation, L.L. and W.L.; writing, W.L. and L.L.; resources, W.L. and Y.W.; review, Y.W. All authors have read and agreed to the published version of the manuscript.

**Funding:** This work was supported by the National Natural Science Foundation of China (Grant no. 12132015).

**Conflicts of Interest:** There are no conflicts of interest regarding the publication of this paper.

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
