# Peer review of "Pressure Drop and Particle Settlement of Gas–Solid Two-Phase Flow in a Pipe"

_applsci, doi:10.3390/app12031623_

Round 1

Reviewer 1 Report

In this work, the authors investigated the particle settlement and pressure drop behaviors in a circular pipe. Transport of particles in a circular pipe can be seen everywhere in various industries, therefore, this work is of importance to characterize the pressure drop and particle settlement. 

However, I am not convinced that how the authors have compared the experimental results in the ref 17. with the numerical simulation. In the ref 17, I found no such experimental values suggested in Fig. 2 and Fig. 3 of the current manuscript. 

The authors should compare their simulation results with well-established experimental data to validate their simulation methods.

Reviewer 2 Report

A review for a manuscript 1553708, titled “Pressure drop and particle settlement of gas-solid two phase flow in a pipe” by Lin et al. to be considered for publication at Applied Sciences 1/17/2022

The authors investigated the particle settlement and pressure drop of gas-solid two-phase pipe flow with varying the inlet velocity, particle volume concentration, mass flow rate, and particle diameter. They derived a momentum equation describing the flow and solved it numerically to study the effects of the above-mentioned conditions on the flow behaviors.

The analysis seems reasonable although there is no interestingly new result, which seems straightforward and can be expected. As long as the journal editor accepts this, I recommend publication but I suggest the following to improve the manuscript.

  1. Page 3 line 109: Any brief descriptions and references for the IPSA_FULL method are recommended. Additionally, any justifications why this particular method is chosen for modeling this system are also recommended.
  2. Page 7 line 253: I understand that those just fitting parameters. However, if any physical meanings are interpreted for those, that would be more meaning.
  3. Page 8 Figure 13 and 14: why do those simulation results have such fluctuations although those are not experimental data? Please explain.
  4. The authors may emphasize the nobility of this work.

Reviewer 3 Report

The article titled "Pressure drop and particle settlement of gas-solid two phase flow in a pipe" evaluated the particle settlement and pressure drop of gas-solid two phase flow in a pipe. The numerical method itself is appropriately validated, and the results are reasonable. Furthermore, the expressions of settlement index, pressure drop per unit length as a function of mixture inlet velocity, particle volume concentration, particle mass flow and particle diameter are valuable data for readers. However, I have several concerns that need to be addressed before considering publication.

  • In section 3.1, what is the IPSA_FULL method?
  • In line 141, you mentioned, "The distinction between the three areas is more obvious with the increase of particle volume concentration.", but the trend of these areas is more evident in the case of low volume concentration for me. Please describe more details.
  • In section 4.4, your definition of αsb and αsu is ambiguous. Please show the exact position in the pipe.
  • In figure 6, please show the colour contour.
  • I think that the settlement occurs mainly due to gravity. How does the advection effect the settlement index? How does the concentration field change if the inlet velocity is zero?
  • In equation 9, η and ζ have dimensions.
  • What is the physical meaning of η and ζ? 

Reviewer 4 Report

Aiming at the gas-solid two-phase flow in a circular pipe, the authors use numerical methods to calculate the momentum equation of the two-fluid model, and compare the results with the experiments. The theoretical results are in good agreement with the experiments. The influences of inlet velocity, particle volume concentration, particle mass flow, particle diameter on particle settlement and pressure drop are discussed, and the correlation expression is given, which has a good summary and innovation. However, there are still some issues that need to be corrected before this paper can be accepted.

  1. There are some redundant citations in the “Introduction”. For example, “particle elasticity” is not studied in this paper, so the citation of "Vasquez et al. [14]..." can be removed.
  2. In Section 4.1,the author mentioned: "The distinction between the three areas is more obvious with the increase of particle volume concentration." However, in Figure 4, it is clear that the distinction between the three areas in the red curve () is more obvious, so Is there a problem with the formulation?
  3. In Figure 4, the pressure distribution decreases along the flow direction, and the pressure drop decreases as the particle phase concentration increases. Can you give a brief explanation for the reasons for these phenomena?
  4. In Figure 5, what is the physical quantity represented by the ordinate? If the ordinate is 2r/D mentioned in the text, should its range be -1.0 ~ 1.0? In addition, the deviation of the maximum velocity and the asymmetry of the velocity are not intuitively represented in the figure, so can the author change the graphic method or give special marks in the figure.
  5. According to the definition of , when is fixed, the more serious the particle sedimentation is, the larger the value of   So why did the author come to the conclusion that "It can be seen that  decreases with the increase of , i.e., the settlement phenomenon of particles is more serious..."?
  6. There is an error in the title of section 4.5, "the settlement index" should be changed to "pressure drop".

Round 2

Reviewer 1 Report

This work well investigates the gas-solid two-pahse flow in a circular pipe numerically based on the momentum equation. 

It is very obvious that the particle settlement becomes more serious with the decrease of V, alpha_s and mass_s, and d_p based on the linear momentum of the particles.  I think all the simulation results are understandable and predictable.

One missing information lacking is, what would happen if particle-to-particle interaction is existing which is not considered in the model.  The authors should at least explain some possible ideas or predictions how to include particle-to-particle interaction in the momentum equation for a future work.
